# iTRAQ-Based Quantitative Proteomic Analysis of Antibacterial Mechanism of Milk-Derived Peptide BCp12 against *Escherichia coli*

**DOI:** 10.3390/foods11050672

**Published:** 2022-02-24

**Authors:** Kun Yang, Yanan Shi, Yufang Li, Guangqiang Wei, Qiong Zhao, Aixiang Huang

**Affiliations:** College of Food Science and Technology, Yunnan Agricultural University, Kunming 650201, China; kunyang128611@126.com (K.Y.); yananshihaha@126.com (Y.S.); yufangfangli@126.com (Y.L.); guangqiangwei@126.com (G.W.); zhaoqiong713@126.com (Q.Z.)

**Keywords:** antimicrobial peptide BCp12, *Escherichia coli*, cell membrane damage, proteomic analysis

## Abstract

BCp12 is a novel casein-derived antibacterial peptide with a broad-spectrum antibacterial effect. However, its action mechanism against *E. coli* is unknown. In this study, the growth curve showed that BCp12 had excellent antibacterial activity against *E. coli*. Red (propidium iodide staining) and green (fluorescein isothiocyanate staining) fluorescence signals were detected at the edges of the *E. coli* cells treated with BCp12. scanning electron microscopy (SEM) and transmission electron microscopy (TEM) images showed that *E. coli* cells became rough and shrunken, and part of the cell contents leaked to form a cavity. Furthermore, the iTRAQ proteome analysis showed that 193 and 174 proteins were significantly up-regulated and down-regulated, respectively, after BCp12 treatment. Four enzymes involved in fatty acid degradation of *E. coli* were down-regulated, disrupting the synthesis of cell membranes. Molecular docking and gel retardation assays showed that BCp12 could bind to genes encoding four key enzymes involved in the fatty acid degradation pathway through hydrogen bonding and hydrophobic interactions, thus significantly inhibiting their activities. Overall, the results indicate that BCp12 inhibits the growth of *E. coli*, causing metabolic disorders, thus destroying the structure of cell membranes.

## 1. Introduction

*Escherichia coli* is one of the most common pathogens causing foodborne diseases [1]. It can easily contaminate food during processing and circulation, especially meat products [2]. For instance, 18.37%, 16.48%, and 46.4% of *E. coli* contamination have been detected in meat products [3], takeout meals [4], and cooked meat products [5], respectively, in Yunnan Province, China. Chemical preservatives, such as potassium sorbate and benzoates, are used to prevent the growth of *E. coli* in foods [6]. However, frequent or excessive use of chemical preservatives can lead to microbial resistance, accumulation of chemical preservative residues, and change in food quality [7]. Therefore, scholars worldwide have mainly explored drugs with strong antibacterial activity against *E. coli* [8].

Antimicrobial peptides (AMPs), as natural active peptides [9], have been recognized as promising candidates to address the challenges of limited safety of chemical food preservatives and antibiotic resistance of multidrug-resistant bacteria [10]. Particularly, AMPs derived from milk have become a research hotspot due to their low toxicity, structural diversity, and diverse bactericidal mechanisms of action [11]. In recent years, several studies have found that AMPs can specifically bind to the negatively charged bacterial membrane, interfering with the hydrophobicity of cells and the integrity of cell membranes [12,13]. For instance, Turk et al. reported that membrane deformation and structural changes in cell membrane lipids lead to bacterial cell death [14]. Moreso, Li et al. reported that the antibacterial actions of ε-PL against *E. coli* occur through morphological damages and cell metabolism inhibition [15,16]. Furthermore, some AMPs can affect the synthesis, expression, and metabolism of bacterial macromolecules through non-membrane permeabilization mechanisms [17], including inhibition of peptidoglycan synthesis [18], direct degradation of target intracellular functions, DNA binding [19], and inhibition of protein synthesis [20]. As a result, it is difficult to clarify the complex and diverse bactericidal mechanisms of antimicrobial peptides. From a technical perspective, most methods (electron microscopic imaging, dye leakage, and gel retardation assays) used to describe the killing mechanisms of AMPs cannot detect the actual interacting components or binding sites. Comparative quantitation of proteomes has become a useful approach for analyzing the underlying mechanism of antimicrobial agents by detecting the differential abundance of proteins. As a result, various studies have utilized the iTRAQ-based quantitative proteomic technology due to its high sensitivity [21,22].

Previous research has demonstrated that BCp12 is a hydrophobic cationic AMP with high thermal stability (primary structure, YLGYLEQLLRLK, and molecular weight, 1508.82 Da) [23]. BCp12 also has a broad-spectrum of antimicrobial activities against foodborne bacteria, including *E. coli*, *Listeria monocytogenes*, *Salmonella typhimurium*, and *S. aureus*. Besides, BCp12 has minimum inhibitory concentration (MIC, 2 mg/mL) of AMPs against *E. coli*. Although BCp12 has antimicrobial spectrums, physicochemical properties, haemolytic properties, hydrophobicity, and surface potential [24], its antibacterial mechanism against *E. coli* is unknown.

In this study, iTRAQ-based quantitative proteomics and ultrastructural observations using scanning electron microscopy (SEM), transmission electron microscopy (TEM) and confocal laser scanning microscopy (CLSM) were used to investigate the antibacterial mechanism of BCp12 against *E. coli* at the proteome level. Multiple experiments were further employed for validation. Molecular docking and gel retardation assays were used to verify several differentially expressed proteins. This study can provide significant insights into the molecular mechanism of BCp12 against *E. coli* pathogens, demonstrating its potential applications in food safety.

## 2. Materials and Methods

### 2.1. Materials and Chemicals

The antibacterial peptide BCp12 was a self-developed sinomenine rennet. A single chromatographic peak was obtained based on living bacteria affinity adsorption and RP-HPLC purification after enzymatic hydrolysis of betelnut river buffalo milk. The chromatographic peak contained four peptide segments. The target active peptide BCp12 was determined through activity screening. Its amino acid sequence was YLGYLEQLLRLK, an antibacterial peptide ɑs1 casein-derived from antimicrobial peptides and synthesized by Guoping Pharmaceutical Co., Ltd. (purity, over 95%).

*Escherichia coli* (*E. coli*) CICC 10003 was obtained from the China Center of Industrial Culture Collection. The bacterial strains were put in glycerol with nutrient broth at −80 °C and were sub-cultured twice in lysogeny broth (LB) medium while agitating (120 rpm) at 37 °C for 24 h before use. The initial concentration of the cells was adjusted to 10^6^ CFU/mL via dilution in phosphate-buffered saline (PBS, 20 mM, pH 7.4) for subsequent experiments.

### 2.2. Growth Curves of E. coli

About 1/2 MIC BCp12 was added to the cell culture. The effects of BCp12 on growth curves of *E. coli* cells were investigated using the modified method reported by Su [25]. Sterile water and tetracycline solution (1 mg/mL) were used as negative and positive controls, respectively. The controls were incubated in a shaker at 37 °C and 120 r/min for 24 h. Samples were taken every 2 h. A microplate reader was used to measure the OD600 value. The process was repeated thrice at each time point, and the average values were used to draw a growth curve.

### 2.3. Confocal Laser Scanning Microscopy (CLSM) Assay

The intracellular localization of FITC-labeled peptides was investigated using a CLSM [26]. The overnight-activated *E. coli* were diluted to 10^6^ CFU/mL, and FITC-BCp12 (final concentration of 1MIC) was added and cultured in a 1.5 mL centrifuge tube at 37 °C for 12 h. The bacteria were then washed thrice using PBS, and the bacterial precipitates were collected. PI (50 µg/mL) was added and incubated in an icebox for 15 min, then washed thrice using PBS, and the bacterial precipitation was collected. A Leica AOBS confocal and scanning laser microscope was used for visualization at 488 nm, 563 nm, and 271 nm. The images were obtained from 500 nm to 530 nm (green channel) and from 535 to 615 nm (red channel) [27].

### 2.4. Determination of the Morphologies of E. coli

*E. coli* suspensions (about 10^6^ CFU/mL) were prepared and separately treated with 0.5× and 1× MIC BCp12. All samples were incubated at 37 °C for 12 h, and the suspensions were centrifuged at 10,000 rpm, 4 °C for 5 min. The precipitated cells were washed thrice and resuspended with PBS. The bacterial precipitate was fixed with 2.5% glutaraldehyde for 6 h. The fixing solution was subsequently removed via centrifugation. The precipitate was again washed thrice with PBS and dehydrated with ethanol standard solutions in a concentration gradient (30%, 50%, 70%, 90%, and 100%) for 10 min (with each ethanol solution in order of increasing concentration). The precipitate was then dried at 60 °C for sputter coating. Cells treated as described above and not exposed to BCp12 were considered as negative controls. A scanning electron microscope (SIGMA 500/VP, Carl Zeiss, Germany) was used for visualization [28].

### 2.5. Transmission Electron Microscopy (TEM) Assay

*E. coli* was diluted to 10^6^ CFU/mL, then BCp12 was added into the fresh cell suspension to a final concentration of 0.5× and 1× MIC, and incubated at 37 °C for 12 h. The treated cells were centrifuged at 10,000 rpm for 5 min, transferred into Eppendorf tubes, and washed thrice using PBS. The resulting cell pellets had a thickness of ≤2 mm and were fixed with glutaraldehyde at 4 °C for 12 h. The fixed pellets were oxidated by adding 200 µL of 1% osmium tetroxide at 4 °C for 2 h. The bacterial pellets were then washed thrice using PBS and sequentially dehydrated using 30%, 50%, 70%, and 90% acetone for 10 min each. The samples were then dehydrated thrice using 100% acetone for 15 min. Finally, the dehydrated pellets were put into the embedding agent for 72 h. The specimen blocks were hand-trimmed using a razor blade and sectioned using an ultra-microtome (Leica EM UC7, Berlin, Germany). The sections were stained with uranyl acetate for 30 min and incubated in lead citrate for 10 min. Cells treated as described above and not exposed to BCp12 were used as negative controls. Transmission electron microscopy (JEOL-JEM-1200 EX, Tokyo, Japan) was used for visualization [29].

### 2.6. Proteome Analysis of E. coli

#### 2.6.1. Total Protein Extraction

The protein was extracted using the method described by Ma [30]. *E. coli* (10^6^ CFU/mL) was cultured in LB broth, then treated with 1 MIC BCp12 (the control was sterile water) at 37 °C for 14 h. The bacterial cell pellets were collected via centrifugation (10,000× *g* at 4 °C for 5 min) and ground into a fine powder in liquid nitrogen. The powder was suspended in lysis buffer (1% sodium deoxycholate (SDS), 8 M urea) with protease inhibitor to inhibit protease activity. The mixture was allowed to settle at 4 °C for 30 min while vortexing the sample every 5 min. The sample was treated via ultrasound at 40 kHz and 40 W for 2 min. The sample was centrifuged at 16,000× *g*, 4 °C for 30 min. The BCA protein assay kit (Pierce, Thermo, Fort Pierce, FL, USA) was used to determine the concentration of protein supernatant via the bicinchoninic acid (BCA) method following the kit protocol.

#### 2.6.2. Protein Digestion and iTRAQ Labeling

Protein digestion was performed via the standard procedure. The resulting peptide mixture was labeled using the plex iTRAQ 8-plex iTRAQ reagent (Applied Biosystems, 4390812), following the manufacturer’s instructions [31]. Briefly, total protein (100 μg) from each sample was mixed with 100 μL of the lysate. TCEP (10 mM) was added and stored at 37 °C for 60 min. Iodoacetamide (40 mM) was then added and stored in the dark at room temperature for 40 min.

Six volumes of cold acetone were added to precipitate protein at -20 °C for 4 h, then centrifuged at 10,000× *g*, 4 °C for 20 min. The pellet was then resuspended with 100 µL and 50 mM triethylammonium bicarbonate (TEAB) buffer. Trypsin was then added at 1:50 (trypsin-to-protein mass ratio) and incubated at 37 °C overnight. One unit of iTRAQ reagent was thawed and reconstituted in 50 µL acetonitrile, after incubating at room temperature for 2 h. Hydroxylamine was added at room temperature for 15 min. The samples were labeled as A1-113, A2-114, A3-115, B1-116, B2-117, and B3-118. Finally, all samples were pooled, desalted, and vacuum-dried.

#### 2.6.3. High pH-RPLC Separation

ACQUITY ultra performance liquid chromatography (Waters, Fort Pierce, FL, USA) with ACQUITY UPLC BEH C18 Column (1.7 µm, 2.1 mm × 150 mm, Waters, Fort Pierce, FL, USA) was used to fraction the pooled samples to increase proteomic depth. Briefly, peptides were separated using an elution gradient (Phase B: 5 mM ammonium hydroxide solution containing 80% acetonitrile, pH 10) at a flow rate of 200 μL/min for 48 min. Twenty fractions were collected from each sample, then pooled, obtaining ten fractions per sample.

#### 2.6.4. LC-MS/MS Analysis

An online nanoflow liquid chromatography-tandem mass spectrometry was used to assess the labeled peptides via the 9RKFSG2_NCS-3500R system (Thermo, Fort Pierce, FL, USA) connected to a Q Exactive Plus quadrupole orbitrap mass spectrometer (Thermo, Fort Pierce, FL, USA) through a nanoelectrospray ion source [32]. Briefly, the C18-reversed-phase column (75 μm × 25 cm, Thermo, Fort Pierce, FL, USA) was equilibrated with solvent A (A:2% ACN with 0.1% formic acid) and solvent B (B: 80% ACN with 0.1% formic acid). The peptides were eluted as follows: 0–63 min, 5–23% B; 63–82 min, 23−29% B; 82–90 min, 29−38% B; 90−92 min, 38−48% B; 92–94 min, 48–100% B; 94–120 min, 100–0% B at a flow rate of 300 nL/min.

The Q Exactive Plus was operated in the data-dependent acquisition mode (DDA) to switch between full-scan MS and MS/MS acquisition automatically. The orbitrap was used to obtain full-scan MS spectra (*m*/*z* 350–1500) with a 60,000 resolution. The automatic gain control (AGC) target was 3e6, and the maximum fill time was 20 ms. The top 20 most intense precursor ions were selected as collision cells for fragmentation using higher-energy collision dissociation (HCD). The MS/MS resolution automatic gain control (AGC) target, maximum fill time, and dynamic exclusion were 15000 (at *m*/*z* 100), 5e4, 45 ms, 45 ms, and 20 **s**, respectively.

#### 2.6.5. Protein Identification

The Proteome Discoverer (Thermo Scientific, Version 2.4) was used to assess the RAW data files in the *E. coli* database (https://www.ncbi.nlm.nih.gov/genome/167, accessed on 19 May 2021). The MS/MS search criteria were as follows: Mass tolerance of 10 ppm for MS and 0.02 Da for MS/MS tolerance, trypsin was used as the enzyme with two missed cleavages allowed, carbamide methylation of cysteine and the ITRAQ of N-terminus and lysine side chains of peptides were fixed modifications, and methionine oxidation was the dynamic modifications. FDR ≤ 0.01 was considered a false discovery rate (FDR) of peptide identification. A minimum of one unique peptide identification was used to support protein identification.

### 2.7. Molecular Docking Analysis

Target proteins (3-ketoacyl-CoA thiolase, acyl-CoA dehydrogenase, S-(hydroxymethyl) glutathione dehydrogenase, acyltransferase) in the fatty acid degradation pathway that was significantly down-regulated were selected as potential target proteins of BCp12 in molecular docking analysis according to KEGG pathway enrichment analysis (*p <* 0.01). The molecular docking software AutoDock Vina 1.1.2 was used to simulate the interaction between BCp12 and target protein [33]. Schrödinger’s Maestro molecular modeling package (Schrödinger, LLC, New York, NY, USA) was used to construct and optimize the structure of BCp12 (YLGYLEQLLRLK). The crystal structures of the target protein were downloaded from the Protein Data Bank (PDB) and Alphfold2 database (https://alphafold.ebi.ac.uk/, accessed on 15 December 2021). All receptor proteins were optimized with PyMOL 2.5 before docking, including removing water molecules, salt ions, and small molecules. Second, PyMOL plug-in Centerof_mass.Py was used to calculate the protein centroid and define the center of the docking box to ensure the docking box wrapped the protein site. The ADFRsuite 1.0 was used to convert all processed small molecules and receptor proteins into pdbqt files. The detail of the global search was set to 20 during docking, while the other parameters remained at default settings. The docking conformation with the highest score was used as the combined conformation by default. Finally, the software PyMOL 2.5 was used for visual analysis.

### 2.8. Gel Mobility Shift Assay

Gel mobility shift assay was performed as described by Bandyopadhyay [34] with slight modifications. Briefly, *E. coli* cells were prepared as described earlier. Bacterial genomic DNA (gDNA) was extracted using an appropriate DNA extraction kit (Soleibao Biotechnology Co., Ltd., Beijing, China). An equal volume of BCp12 solution at different concentrations (0, 1, and 2 mg/mL) was added to 5.0 μL of gDNA, then incubated at 37 °C for 30 min. Electrophoresis was performed on agarose gel (1% mass fraction). DNA migration was visualized under UV illumination using a gel imaging system (Bio-Rad, Hercules, CA, USA).

### 2.9. Statistical Analysis

All experiments were performed in triplicate. Data are presented as a mean value and standard deviation (mean ± SD). GraphPad Prism 8.0.1 and Adobe Illustrator CS6 were used to draw graphs.

## 3. Results

### 3.1. The Effect of BCp12 on E. coli Growth

The inhibitory effects of BCp12 on *E. coli* are shown in Figure 1A. *E. coli* cells grew into stationary phases after 14 h, and the OD600 value reached 0.8. However, 1/2 MIC BCp12 partially inhibited *E. coli* growth, and the OD600 value reached 0.7 after a 24-h treatment. Moreover, tetracycline significantly inhibited the growth of *E. coli* (*p* ≤ 0.001). These results indicate that BCp12 can inhibit the growth of *E. coli*. The integrity of the cell membrane was measured to determine the underlying inhibition mechanism of the growth of *E. coli* after BCp12 treatment.

### 3.2. Localization Analysis of BCp12 in E. coli Cells

PI and FITC experiments were conducted to assess if the outer membrane of *E. coli* was seriously damaged. Red-fluorescent PI is impermeable to the cell membrane and enters cells through compromised membranes, thus staining the nucleic acids [35]. Red (Figure 1(Ba,Bc)) and green (Figure 1(Bb,Bd)) fluorescence signals were detected at the edges of the *E. coli* cells treated with BCp12 (MIC) for 12 h, indicating that the integrity of cell membranes was severely damaged, allowing the entry of FITC-BCp12 and PI (Figure 1B). Benincasa found that antimicrobial peptides first bind to the cell membrane surface of *E. coli* and then enter the cell membrane due to their amphiphilic structure to form short-lived pores [17]. Some peptides are transferred to the lobules in the membrane when these pores are damaged [36], and some cells treated with FITC-F1 conjugate produce yellow-green fluorescence [37]. Furthermore, this study proved that FITC-BCp12 could bind to the cell membrane surface of *E. coli* and then enter the cell membrane because the integrity of the cell membrane of BCp12-treated *E. coli* was severely damaged.

### 3.3. Effect of BCp12 on E. coli Morphologies

There were significant differences between the control and 1/2 MIC, 1 MIC treatment groups after 12 h of incubation (Figure 1C). SEM (Figure 1(Ca)) showed that the cells in the untreated *E. coli* had a smooth, intact, and plump structure. However, *E. coli* cells showed a plump profile with pustule-like bulges and folds on the surface of the cells after treatment with 1/2 MIC BCp12 (Figure 1(Cb)). The cells treated with 1 MIC BCp12 (Figure 1(Cc)) were significantly deformed and had tears and holes on the surface. The “blanket” model suggests that once the antimicrobial peptide binds to the surface of the bacterial cell membrane, it is covered by a ‘carpet’, forming an instant hole. The hole subsequently increases the membrane permeability and causes the cell to collapse [38]. Meanwhile, Brogden also found that the blanket model does not require the special structure of AMPs and forms membrane pores; however, the blanket model relies on the positive charge of AMPs to be distributed across the entire polypeptide sequence [39]. The amino acid residue of BCp12 is YLGYLEQLLRLK conforms to the residue distribution pattern of the antimicrobial peptide in the blanket pattern. Therefore, BCp12 could have destroyed the cell membrane structure of *E. coli* through the “blanket mode”.

### 3.4. The Effect of BCp12 on the Internal Structure of E. coli Cells (TEM Observation)

As shown in Figure 1(Da), the normal control cells had a fluffy boundary with intact cell wall and membrane structures, and the cell content was substantial and dense. However, the cell membranes had holes, tears, and injuries after BCp12 treatment for 12 h (Figure 1(Db)) and (Figure 1(Dc)). Besides, there was leakage of intracellular materials, and some cell wall membranes, organelles, and nucleus areas disappeared. Chang demonstrated that OPh-PANF could damage the cell membrane structure of *E. coli* after 6 h of treatment, making the cell appear as a vacuole structure [40]. Yang et al. also found that AMPs could rupture the cell membrane, leading to the leakage of intracellular material, thus causing cell death [41,42]. In this study, BCp12 destroyed the cell membrane. Furthermore, BCp12 could have inhibited the growth of *E. coli* due to the damaged cell membrane.

### 3.5. Proteome Analysis

#### 3.5.1. Protein Extraction and Basic Identification Information

SEM and TEM observation showed that BCp12 damaged the cell membranes of *E. coli*, and iTRAQ proteome analysis was conducted to assess the mechanism of cell membrane damage. The SDS-PAGE results showed that the band distribution and grayscale were the same in the three biologically duplicated protein samples with different treatments (Figure 2A), indicating that the sample was well reproduced from the bacterial culture to protein extraction. Mass spectrometry detected 979,799 spectra in *E. coli*. The obtained spectra were matched to the protein database of *E. coli*, and 23,469 peptides were identified. Assembly and screening of the peptides were conducted, and 2595 proteins were identified as belonging to the proteome of *Escherichia coli* (Figure 2B). The relative molecular masses of all the identified proteins were mainly between 1 and 201 kDa. A total of 1076 proteins had molecular weights between 21 and 41 kDa, with only one protein having 180–201 kDa (Figure 2C).

#### 3.5.2. LC-MS/MS Analysis

The quantitative values with missing data were removed after normalization and then subjected to cluster analysis (Figure 3A). The results showed good reproducibility between the three biological replicates of the BCp12 treatment group (A1, A2, A3) and the control group (B1, B2, B3). The protein abundance between the control group and the BCp12-treated group was also significantly different.

The volcano map of the differentially expressed proteins of *E. coli* was also analyzed (Figure 3B) (X-axis is log2 fold-change). The thresholds of fold-change (>1.2 or <0.83) and p-value < 0.05 were used to identify the differentially expressed proteins. The light blue dots on the upper left indicate significantly down-regulated proteins, and the dark blue dots on the upper right show significantly up-regulated proteins. A total of 193 proteins were significantly up-regulated, and 174 proteins were significantly down-regulated after BCp12 treatment.

#### 3.5.3. Functional Annotation of Differentially Expressed Proteins

Differentially expressed proteins of *E. coli* (fold-change ≥ 1.2 times) were used for Gene Ontology (GO) annotation (http://www.blast2go.com/b2ghome; http://geneontology.org/, accessed on 19 May 2021) (Figure 3). The cellular component analysis (Figure 3C) showed that the protein-containing complex was the most down-regulated. Similarly, the cellular anatomical component of *E. coli* was the most up-regulated, as shown by pustule-like bulges in SEM (Figure 1C). The molecular functional analysis (Figure 3D) showed that BCp12 treatment had significant binding, catalytic activity, transporter activity, and transcription regulator activity effects. The biological process was annotated using GO Slim to analyze gene functions at a macroscopic level (Figure 3E). BCp12 treatment significantly affected the cellular process, followed by the metabolic process. BCp12 treatment also down-regulated the biological adhesion and multi-organism processes.

#### 3.5.4. Expression Changes of Cell Membrane-Related Proteins

Four enzymes involved in fatty acid degradation of *E. coli* were significantly down-regulated after BCp12 treatment (Figure 4) by <0.83 folds (fadA; −0.470927 times, fadE; −0.350209 times, frmA; −0.286683 times and aas; −0.337095 times).

Fatty acids are the main components of cell membranes and also precursors of cell envelope biosynthesis [43]. However, β-oxidation can degrade excess fatty acids or intermediate products of *E. coli* as follows: fatty acyl-Co A conversion to enoyl-Co A via fade [44]. Besides, fadE is the only acyl-Co A dehydrogenase in *E. coli* [45]. Knocking out the endogenous fadE gene, which encodes an acyl-CoA synthetase, blocks fatty acid degradation [46]. Some studies have also shown that acetyl-CoA C-acyltransferase fadI and acetyl-CoA C-acyltransferase fadA are significantly down-regulated in BM1157-treated *E. coli* [47]. Moreso, a loss of frmA increases cell-cell adhesion [48]. Therefore, damage to the *E. coli* cell membrane may be due to the interaction between BCp12 and the fadE gene, which decreases fadE enzymatic activity and blocks fatty acid degradation, thereby hindering cell membrane synthesis.

### 3.6. Molecular Docking Analysis

The AlphaFold Protein Structure Database (AlphaFold DB, https://alphafold.ebi.ac.uk, accessed on 15 December 2021.) is an openly accessible and extensive database that can be used to accurately predict protein structure [49,50]. In this study, the crystal structures of target protein (3-ketoacyl COA thiolase, s-(hydroxymethyl) glutathione dehydrogenase) were downloaded from the Alphfold2 database (https://alphafold.ebi.ac.uk/, accessed on 15 December 2021). Molecular docking experiments were then performed to prove the above hypothesis. BCp12 formed hydrogen bonds with residues ala-174 and pro-133 and hydrophobic interaction with tyr-178, phe-394, val-67, gln-69, pro-133, pro-41, val-136 and val-171 in the active site of 3-ketoacyl-CoA thiolasecan (Figure 5A). BCp12 also formed hydrogen bonds with residues arg-483, asn-429, gly-428, ser-191, thr-185, Glu-270, pro-220, and arg-416 and hydrophobic interaction with ille-423, pro-419, leu-261, and phe-216 in the active site of acyl-CoA dehydrogenase (Figure 5B). BCp12 formed hydrogen bonds with residues tyr-88, ile-287 and gln-294, and hydrophobic interaction with phe-45, val-291, val-289, ile-296, trp-309, phe-314, leu-105 and ile-287 in the active site of acyl S-(hydroxymethyl) glutathione (Figure 5C). Finally, BCp12 formed hydrogen bonds with residue phe-344, arg-10, ser-78, phe-341, pi-pi conjugate with residues phe-146, and hydrophobic interactions with residues met-85, pro-86, VAL-89, Leu-93, leu-97, phe-146, leu-143, pro-340, and phe-341 in the active site of acyltransferase (Figure 5D). Additionally, the best binding energy of BCp12 to 3-ketoacyl-CoA thiolase, acyl-CoA dehydrogenase, s-(hydroxymethyl) glutathione dehydrogenase and acyltransferase were −6.3 kcal/mol, −6.9 kcal/mol, −7.0 kcal/mol, and −7.5 kcal/mol, respectively (Table 1). The binding energy was less than −5 kcal/mol, indicating that a stable complex can be formed between ligand and receptor [51].

The scores of the four groups were less than −5 kcal/mol, indicating that BCp12 has a strong binding force with the four key enzymes. BCp12 had a strong inhibitory effect on the four key enzymes, possibly inhibiting the fatty acid degradation pathway.

### 3.7. In Vitro Interaction between BCp12 and DNA

Electrophoretic mobility of DNA coupled with BCp12 was measured to determine whether BCp12 targets DNA. BCp12 could bind to *E. coli* DNA in the untreated control to a certain extent, thus forming a bright electrophoretic product (Figure 6). A certain amount of DNA interacted with BCp12 at 1–2 mg/mL BCp12, forming a less visible electrophoretic product. Xie [52] found that antimicrobial peptides bind to the surface of DNA at low concentrations, thus blocking DNA migration. In contrast, high concentrations of antimicrobial peptides bind to the surface of the DNA double helix through electrostatic force, thus competing and hindering the binding of ethidium bromide with DNA, hence forming no electrophoresis product. Yonezawa [53] reported that tachyplesin I exerts its antimicrobial activity by binding to DNA, thus inhibiting the synthesis of macromolecules. Moreover, the antimicrobial activity of HMGN2 against *E. coli* might be associated with its ability to bind to DNA [54]. Therefore, BCp12 can bind to DNA in vitro and affect gene transcription and translation, consistent with expression changes of cell membrane-related protein findings.

## 4. Conclusions

BCp12 treatment damaged the cell envelope integrity of *E. coli*, and some cell contents leaked to form a cavity, making the FITC-BCp12 and PI enter the cells and emit green and red fluorescence signals at the edges. Furthermore, the iTRAQ proteome analysis showed that BCp12 treatment significantly influenced cellular, metabolic, biological adhesion, and multi-organism processes. Importantly, BCp12 could bind to genes encoding four key enzymes involved in the fatty acid degradation pathway through hydrogen bonding and hydrophobic interactions, thus significantly inhibiting their activities, which disrupts the synthesis of cell membranes. Overall, these results indicate that BCp12 can inhibit the growth of *E. coli*, cause metabolic disorders, and destroy the structure of cell membranes. Understanding the bactericidal mechanism of BCp12 can guide its future development and utilization as an effective antibacterial agent.

## Figures and Tables

**Figure 1 foods-11-00672-f001:**
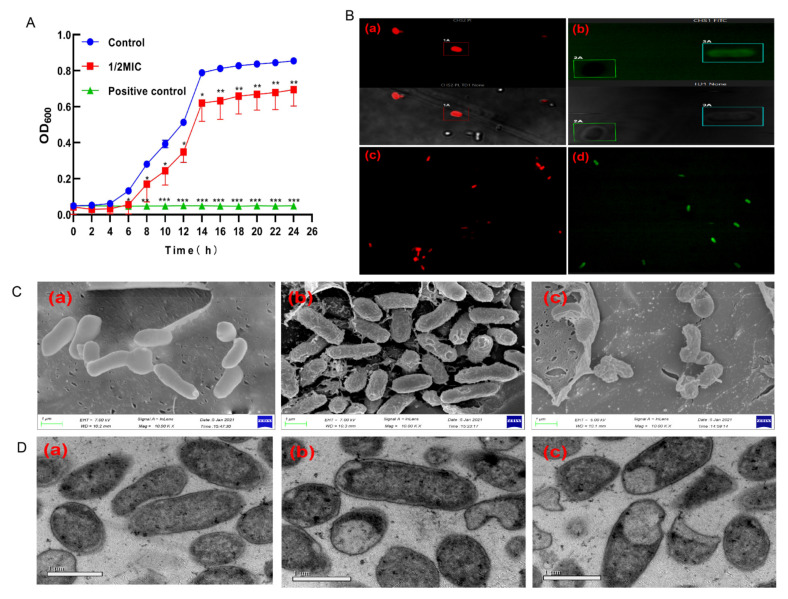
Damage effect of BCp12 to *Escherichia coli* cell membrane. (**A**) Growth curves for *E. coli* cultured in sterile water and 1/2 MIC BCp12. Note: compared with the control group, *, *p*
*≤* 0.05; **, *p*
*≤* 0.01; ***, *p*
*≤* 0.001. Each value represents the average of three independent measurements. (**B**) Effects of BCp12 on cell membrane integrity of *E. coli* via CLSM. The *E. coli* cells treated with 1 MIC BCp12 ((**a**,**b**) represent the bright field and fluorescence image; (**c**,**d**) represents the image under the fluorescent field). (**C**) SEM analysis of *Escherichia coli* under different treatments; (**a**)—the cells in the untreated *E. coli*; (**b**)—the cells after treatment with 1/2 MIC BCp12; (**c**)—the cells treated with 1 MIC BCp12; (**D**) TEM analysis of *Escherichia coli* under different treatments. (**a**) Untreated *E. coli*. (**b**) *E. coli* treated with 1/2 MIC BCp12. (**c**) *E. coli* treated with 1 MIC BCp12.

**Figure 2 foods-11-00672-f002:**
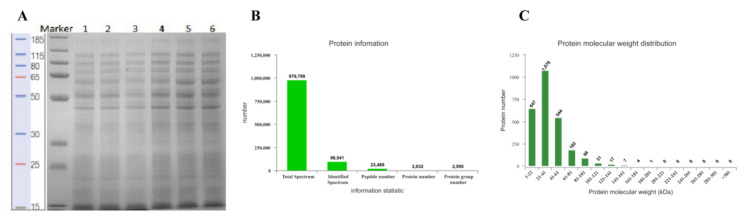
Total protein quantification and basic identification information of *E. coli*. (**A**) SDS-PAGE of *E. coli* proteins; lane 1–3: samples of BCp12 treatment; lane 4–6: samples of 0.1 M PBS buffer treatments (control). (**B**) Protein information. (**C**) Protein molecular weight.

**Figure 3 foods-11-00672-f003:**
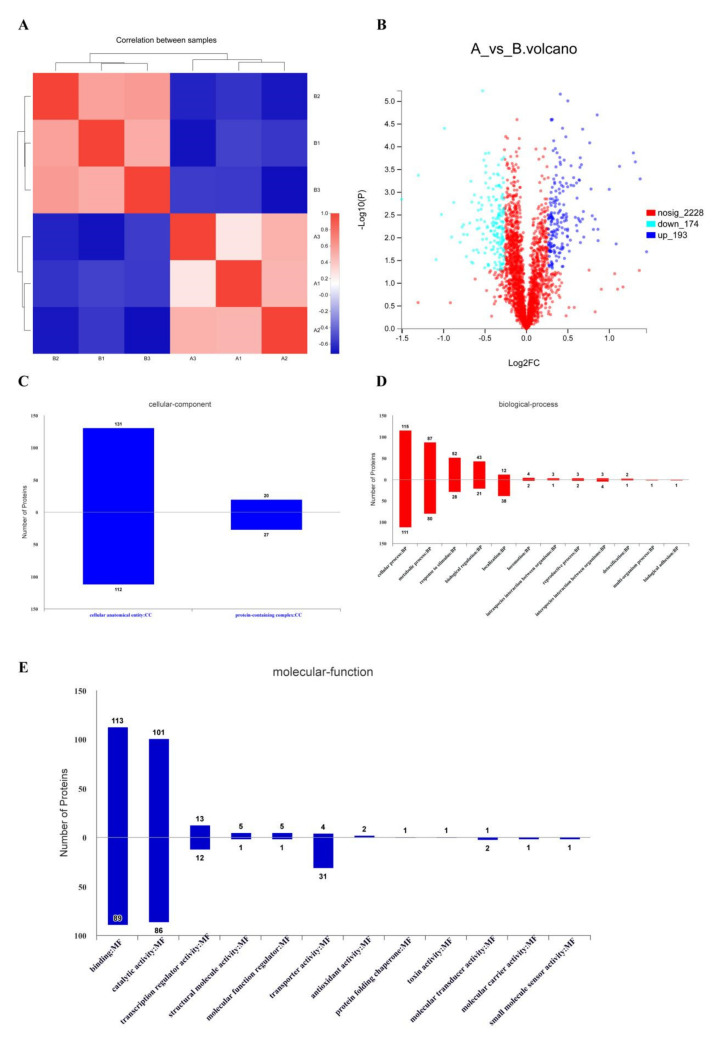
ITRAQ proteome profiling was identified in *E. coli* treated with BCp12. (**A**) Heat map of proteins from *E. coli*; treatments: A1, A2, and A3; control: B1, B2, and B3. (**B**) The volcano map of the differentially expressed proteins in *E. coli*. (**C**) Cellular component of GO analysis. (**D**) Molecular function of GO analysis (**E**) Biological process of GO analysis.

**Figure 4 foods-11-00672-f004:**
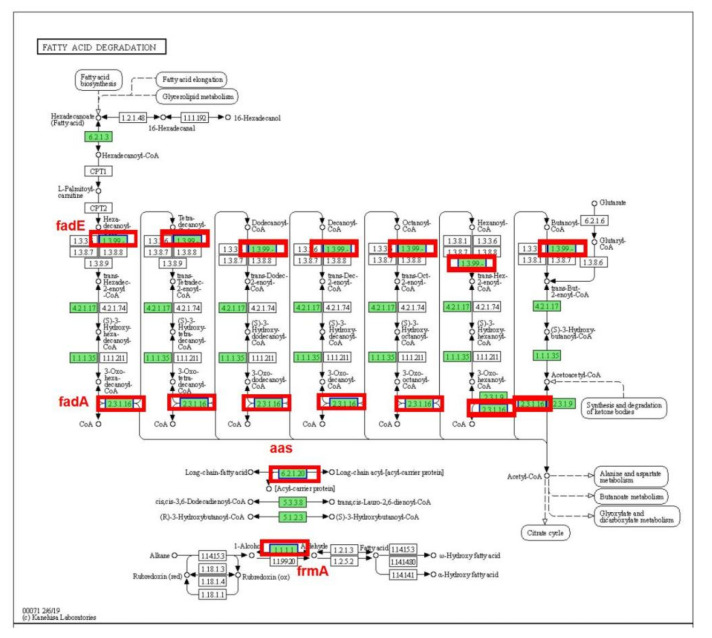
Fatty acid degradation pathway of *E. coli*; Green indicates down-regulation > 0.83 times; and red shows down-regulation < 0.83.

**Figure 5 foods-11-00672-f005:**
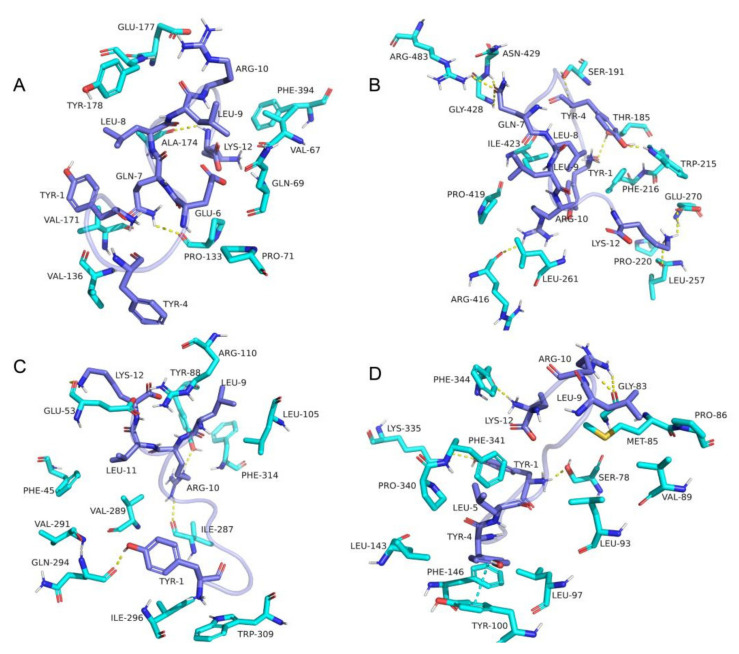
Molecular docking analysis of the binding mode of BCp12. (**A**) 3-ketoacyl COA thiolase (fadA, Alphafold database). (**B**) acyl-CoA dehydrogenase (fadE, PDB ID:3DJL). (**C**) s-(hydroxymethyl) glutathione dehydrogenase (frmA, Alphafold database). (**D**) acyltransferase (aas,PDB ID:6Q3A). The yellow dotted line indicates hydrogen bonding, and the cyan dotted line indicates pi-pi conjugation.

**Figure 6 foods-11-00672-f006:**
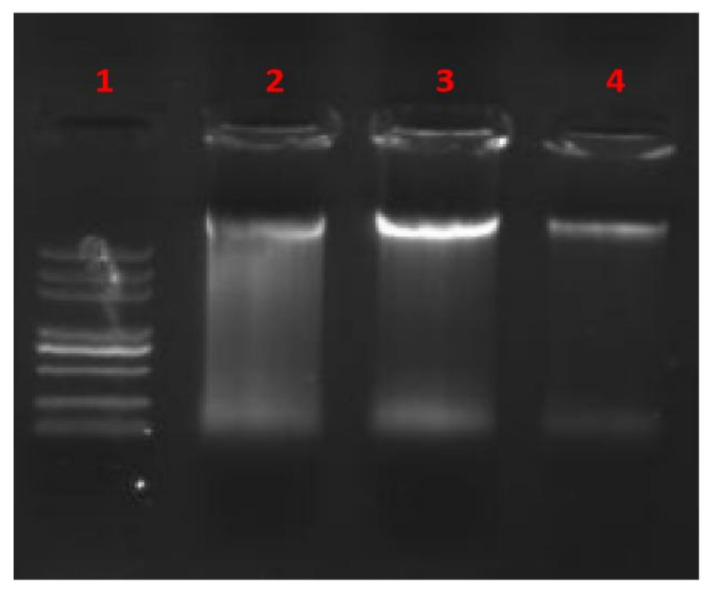
Gel retardation assay of BCp12. (1) Marker; (2–4) BCp12 at concentrations 1, 0, 2 mg/mL, respectively.

**Table 1 foods-11-00672-t001:** Docking binding energy between BCp12 and target protein.

Target Protein	Docking Score (kcal/mol)	Hydrogen Bond Binding Site	Hydrophobic Interaction Binding Site
3-ketoacyl-CoA thiolase (fadA)	−6.3	ala-174,pro-133	tyr-178, phe-394, val-67, gln-69, pro-133, pro-41, val-136 and val-171
acyl-CoA dehydrogenase (fadE)	−6.9	arg-483, asn-429, gly-428, ser-191, thr-185, Glu-270, pro-220 and arg-416	ille-423, pro-419, leu-261 and phe-216
S-(hydroxymethyl)glutathione dehydrogenase (frmA)	−7	tyr-88, ile-287 and gln-294	phe-45, val-291, val-289, ile-296, trp-309, phe-314, leu-105 and ile-287
Acyltransferase (aas)	−7.5	phe-344, arg-10, ser-78 and phe-341	met-85, pro-86, VAL-89, Leu-93, leu-97, phe-146, leu-143, pro-340 and phe-341

## Data Availability

The data presented in this study are available within the article.

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
