# Peer review of "iTRAQ-Based Quantitative Proteomic Analysis of Antibacterial Mechanism of Milk-Derived Peptide BCp12 against Escherichia coli"

_foods, 2022, doi:10.3390/foods11050672_

Round 1
Reviewer 1 Report
I appreciate authors for their efforts to improve the manuscript. Each points are now addressed wisely in the revised manuscript, except Figure 5.
I have only one concern related to Figure 5. To make it more clear, ribbons can be removed or the ligand can be represented by bold sticks. This wire structure of amino acids as well as of ligand is not clear. It's difficult to differentiate amino acids and ligands.
Author Response
View attachments,please!

Reviewer 2 Report
The manuscript is interesting and It is good writing
Author Response
Please see the attachment.

This manuscript is a resubmission of an earlier submission. The following is a list of the peer review reports and author responses from that submission.
Round 1
Reviewer 1 Report
Dear Editor,
The authors conducted experimental studies on “iTRAQ-based quantitative proteomic analysis of antibacterial mechanism of Milk-derived Peptide BCp12 against Escherichia coli”. However, the MS includes interesting studies but could not be accepted based on single technique. Further studies are also required to describe this peptide as an antibacterial agent.
Introduction- It is mentioned that antibacterial mechanism of BCp12 against E. coli is unknown. Has any experiment been carried out to understand antibacterial mechanism of BCp12 against E. coli in this MS?
The significance of using iTRAQ based quantitative proteomic analysis is not clear.
Methodology is too long. It should be shortened by giving suitable references.
In methodology, it is written that the antimicrobial peptide, BCp12, was isolated? How it was isolated remains unclear.
The MS is poorly written.
There are numerous, language, syntax and grammatical errors.
E. coli is not italics throughout the MS
Line 77 adjusted to 106
Line 273 for the inhibited the and so many………………
Reviewer 2 Report
In the control curve of figure 1A the authors should show the bars of the standard deviation. The authors should use a test of t student or fisher to compare the two treatments.
Reviewer 3 Report
The manuscript entitled "iTRAQ-based quantitative proteomic analysis of antibacterial mechanism of Milk-derived Peptide BCp12 against Escherichia coli" is an interesting study pointing out the mechanism of antibacterial activity of Milk-derived Peptide BCp12 against E. coli. The manuscript seems to be of high importance in food and pharmaceutical industry. The experimental details are sufficient. However, the following points need to be addressed before possible acceptance of the manuscript.
Please make "Escherichia coli or E. coli" in italics throughout the manuscript.
Why in the study of the effect of BCp12 on E. coli growth, any positive control (drug active against E. coli) is not included?
Please improve Figure 5, for better visualization. Also, include the PDB ID of the target receptor in the figure legend.
If the structures were not available on PDB site, how the authors modelled the receptor and validated them?
In Table 1, authors can include the information about the binding amino acids that interacts with BCp12, for quick understanding.
